# Specific Neurodynamic Exercises on Pain and Disability in Old Women with Chronic Mechanical Neck Pain: A Randomized Controlled Trial

**DOI:** 10.3390/healthcare12010020

**Published:** 2023-12-21

**Authors:** Luis Polo-Ferrero, David Canchal-Crespo, Susana Sáez-Gutiérrez, Arturo Dávila-Marcos, Ana Silvia Puente-González, Roberto Méndez-Sánchez

**Affiliations:** 1Department of Nursing and Physiotherapy, University of Salamanca, 37007 Salamanca, Spain; pfluis@usal.es (L.P.-F.); susanasg@usal.es (S.S.-G.); turitodavila@usal.es (A.D.-M.); ro_mendez@usal.es (R.M.-S.); 2Institute of Biomedical Research of Salamanca (IBSAL), 37007 Salamanca, Spain; 3Navarra University Hospital, Navarra Health Service, 31008 Pamplona, Spain; dcanchalc@usal.es

**Keywords:** neck pain, exercise, immune system, aging, women and physical therapy

## Abstract

Neurodynamic exercise is a specific type of exercise used as a neural treatment that focuses on restoring altered homeostasis in the neuroimmune system by mobilising the nervous system and other structures. A prospective, randomized clinical trial was performed to evaluate the effect of neurodynamic exercises on disability and neck pain in elderly women over four weeks. Participants were randomized into two groups: a neurodynamic (NM) group (n = 28) and a non-specific exercise (NSE) group (n = 28). Inclusion criteria were women over 65 years of age who subjectively admitted to having mechanical neck pain for more than six months. Results showed that specific neurodynamic exercises can improve pain and disability in older women with chronic mechanical neck pain. Improvements were observed in all variables (*p* < 0.05). Significant between-group differences in favour of the NM group were only found for neck pressure pain thresholds and both tibialis anterior muscles. Larger effect sizes were obtained in favour of the NM group, especially for pain, disability, neck extension and inclination and pressure pain thresholds. Neurodynamic exercises have been shown to be more clinically relevant in disability and neck pain in older women.

## 1. Introduction

Neck pain represents a prevalent musculoskeletal disorder, with a standardised prevalence rate of 27 cases per 1000 people in 2019 [1]. Healthcare expenditure statistics from 2016 reveal that low back and neck pain ranked as the highest outlay, with an estimated expenditure of USD 134.5 billion in the United States [2]. In addition, neck pain is the fourth leading cause of lost years due to disability and is the second most prevalent and second most common musculoskeletal condition among women [1,3]. The burden of pain and the number of years lived with disability were higher in women and increased with age, with ageing being one of the most important risk factors for experiencing pain due to changes in the normal anatomy of the cervical spine [1,4]. Besides gender and age, given the tendency of neck pain to manifest as a chronic problem with a multifaceted aetiology, it is crucial to consider modifiable and non-modifiable risk factors for neck pain [1]. In line with existing evidence on neck pain, we need to consider psychological risk factors, in particular mood disorders [5], as well as stress, distress, anxiety, cognitive functioning and pain-related behaviours, and in addition, biological risk factors, including occupational factors, neuromusculoskeletal disorders, autoimmune diseases and genetic predispositions [1,4]. All of these factors make an individual’s experience of neck pain a multidimensional phenomenon in which physical, psychological and social dimensions influence one another. These findings suggest that health care professionals should be aware of recognizing and assessing all of the individual’s experiences in order to provide a truly patient-centred care pathway [6].

People with neck pain should learn self-treatment methods to improve recurrence rates, avoiding medication, medical visits and associated expenses. Therapeutic exercise is essential, and neurodynamic (NM) exercise is a specific type of exercise used as a neural treatment that focuses on restoring altered homeostasis in the neuroimmune system by mobilising the nervous system and other surrounding structures [7].

The brachial plexus is a group of nerves that run from the lower part of the neck (C5 to T1) to the arm. The main entrapment points of the brachial plexus where the nerves can suffer are the cervical junction foramina, the upper thoracic opening, the scalene triangle, the costoclavicular interval and the loop between the coracoid and the pectoralis minor [8].

Recently, it has been reported that removal of neural traction load may have deleterious effects on the nervous system. The effects of tension and tensile loading help restore homeostasis to the site of entrapment, the dorsal root ganglia, and the spinal cord. In this way, nerve regeneration is favoured, greater muscle power is obtained and there is less hyperalgesia and mechanical and thermal allodynia and a better modulation of conditioned pain [9]. By stimulating the posterior part of the spinal cord, it enables the first level of integration into the central nervous system (CNS), which can lead to changes in the experience of pain [10]. Furthermore, cognitive variables such as kinesiophobia, catastrophization and passive coping must be considered; they can influence the CNS and the person’s painful experience [11,12].

Treatment strategies for neck pain are typically multimodal [13,14]. However, evidence for effective treatment of nerve-related pain is lacking. The positive effects that different manual therapy techniques can have, such as manipulations or the treatment of myofascial trigger points, and the powerful effect that different types of exercises can achieve have been demonstrated [15,16,17]. Furthermore, the possible effect that techniques performed in the cervical region may have on upper limb pain and neuropathic pain has also been demonstrated [18,19]. Nevertheless, there are not many articles demonstrating the effect on pain in general. After the improvement of symptoms in clinical experience, it is hypothesised that NM may be effective in the treatment of neck pain.

## 2. Materials and Methods

### 2.1. Study Design

A prospective, randomized clinical trial was performed over four months to evaluate the effect of four weeks of neurodynamic exercises on disability and neck pain. Participants were enrolled in the study and were randomized into a neurodynamic (NM) group (n = 28) and a non-specific exercise (NSE) group (n = 28), making it a randomized parallel group trial. Before participating in the study, all subjects read and signed the informed consent form. A pre and final assessment was performed, with a four-week intervention with 3 sessions per week. After screening and inclusion of participants, the randomisation process was carried out. All participants were randomly assigned to each group by an independent researcher using Epidat 3.1. Participants were unaware of their assigned intervention, and the evaluating and data-acquiring researchers were blinded. Exercise sessions were conducted by other investigators.

The primary objective was to determine the effect of neurodynamic exercises on disability and neck pain in elderly women compared to an NSE group performing non-specific exercises. Changes in pain, disability, range of motion (ROM), upper limb strength, pressure pain threshold, kinesophobia and catastrophizing were evaluated.

The report of this clinical trial conforms to CONSORT 2010 [20], it received approval from the Salamanca Health Area Drug Research Ethics Committee (code PI 2022 03 973) and it has been registered with clinicaltrials.gov with the code NCT05596019. Both the acceptance by the bioethics committee and the registration of the study in clinicaltrials.gov were carried out prior to starting the clinical trial.

### 2.2. Participants

Participants were introduced to what the study was about, the objectives, how it would be carried out, both the assessments and the intervention. They were told participation would be voluntary and that they could drop out at any time. The procedures followed were in accordance with institutional guidelines.

Inclusion criteria were women over 65 years of age who subjectively admitted to having mechanical neck pain for more than six months, and all participants had to have signed the informed consent form. Exclusion criteria were receiving other physiotherapy treatment in the period, having started taking analgesics or anti-inflammatory drugs 4 weeks previously or having recent cervical trauma/surgery.

### 2.3. Evaluations

The assessments were carried out by a blind evaluator with more than 5 years of experience. The methodology carried out in the different measurements can be seen in Appendix B (Table A1). The following variables were measured:Pain (numeric pain rating scale, NPRS): The NPRS was used to determine the intensity of neck pain at the beginning and at the end of the intervention. Patients were asked to indicate the intensity of their neck pain at the time of the evaluation [21].Cervical disability (neck disability index, NDI): The NDI is the most commonly used current self-report measure for neck pain [22] and is a Spanish validated self-report questionnaire used to determine how neck pain affects the patient’s daily life and to assess the self-assessed disability of patients with neck pain [23].ROM (°): Cervical free joint range was measured using the CROM-3 cervical joint range measuring device (Performance Attainment Associates, St. Paul, MN, USA). It has been shown that ROM measured with a commonly used device does not seem to improve reliability compared to technological devices [24].Hand dynamometry (kg): Hand dynamometry was performed using a Jamar^®^ Plus Hand Dynamometer device (Serial number 2019070288, Paterson Medical, Green Bay, WI, USA), which is considered the gold standard in assessing grip strength [25].Upper limb dynamometry (kg): Upper limb dynamometry was performed using a McroFET™ 2 device (Serial number 1B118W, Hoggan Scientific L.L.C., Salt Lake City, UT, USA) to assess the changes in the motor behaviour of the main metameric levels from which the main nerve trunks depart [26]. Level C5 (Fifth cervical level) was assessed with elbow flexion, C6 with wrist extension and C7 with elbow extension.Pain pressure threshold (kg/cm^2^): Pain pressure threshold was assessed using a Pain Test™ FCMI 25 digital algometer (Serial number J481317, Wagner Instruments, Greenwich, CT, USA). The points were the spinous nerves of C5, C6, C7, both median nerves and both tibialis anterior (TA) muscles to check possible systemic effects [27,28].Kinesiophobia: Kinesiophobia was evaluated using the TAMPA TSK-11 questionnaire, a self-reported questionnaire with 11 items that was developed to measure fear of movement [29].Catastrophizing: Catastrophizing was evaluated using the pain catastrophizing scale (PCS), a self-reported measure of 13 items that assesses the negative and exaggerated mental perception regarding a person’s experience of pain, both real and anticipated [30].

### 2.4. Interventions

Both groups completed 50-min sessions. The sample session was divided into three parts. The initial part and the final part of returning to calm was the same in both groups. Only the main part differed:Neurodynamic group: The NM group exercises are explained in Appendix C (Table A2). They consist of neurodynamic median, ulnar and radial nerve gliding exercises and exercises of the main entrapment points described in thoracic outlet syndrome. The movement exercises were performed with 15 repetitions, 3 times with 30 s breaks. Stretching exercises were performed 1–2 times for 1 min. The TIDIER checklist was completed and is attached in the Appendix A.Non-specific exercise group: The NSE group completed an exercise program that consisted of walking and running at different paces, upper and lower limb strength exercises and balance exercises.

Both programs have been proven to have a scientific basis in improving pain tolerance. A meta-analysis showed that neurodynamics can have an effect on disability and pain in general, not just on neuropathic pain, and nonspecific exercise has been shown to be highly effective in reducing neck pain [31,32].

### 2.5. Statistical Analysis

The sample size was calculated using the GRANMO tool, version 7.12 of April 2012. For people with mechanical neck pain, the minimum clinically important difference (MCID) for NPRS was 1.3 points and 19 percentage points for NDI [33]. Considering MCID for NPRS, the calculation was performed with a bilateral contrast with a risk alpha of 0.05 and a risk beta of 0.2, and a dropout rate of 15%. Subsequently, it was concluded that 25 subjects per group were needed to detect a difference equal to or greater than 1.3 units, assuming a standard deviation of 1.5 units.

Only women who completed the intervention were included in the analysis (per protocol analysis). Descriptive statistics, such as mean and standard deviation, were employed to assess participant characteristics, including their central tendency and variability. IBM SPSS statistical software (SPSS 26 Inc., Chicago, IL, USA) was used for statistical analysis. The distribution was tested for normality in the normality plots and was checked together with the homogeneity of all variables using the Kolmogorov–Smirnov test and Levene’s test, respectively.

Means of variables that were homogeneous and normal were compared using the parametric paired samples Student’s *t*-test. For variables with non-normal and heterogeneous distribution, the non-parametric Mann–Whitney U-test was used. The analysis was performed on the difference of the means of the two assessment moments. The confidence level used was 95% (0.05).

To compare whether the effect of one intervention was superior to that of the other, Student’s *t*-test for equality of means was performed, and the value was taken assuming equal or non-equal variances, depending on whether or not it was analysed by parametric means.

Effect size was analysed by comparing differences between means using Cohen’s d to test whether the effect of the interventions was clinically relevant.

## 3. Results

### 3.1. Baseline Characteristics of the Participants in Both Groups

The study sample consisted of 56 women who met the inclusion and exclusion criteria, distributed into the NM group (n = 28), with a mean age of 76.27 (SD 5.18), and into the NSE group (n = 28), with a mean age of 74.69 (SD 5.03). Finally, 53 women completed the programme. In the NM group, two women had to drop out due to COVID-19 infection, and in the CG group one woman had to drop out due to surgical intervention (Figure 1).

We checked whether there were significant differences in the results when analysing by protocol and by intention to treat. Since only three older women dropped out, only small differences in the means were observed, which did not affect the statistical results obtained in both groups. The per-protocol analysis was chosen to test the effect of neurodynamics in an ideal environment and to minimize bias by excluding people who did not strictly comply with the study protocol.

Table 1 shows the means and standard deviations of all baseline variables of the two groups. Normality and homogeneity values are also shown. The Shapiro–Wilks test showed that all variables had normal values (*p* > 0.05), except for catastrophizing (Sig. = 0.010), algometry in the right median nerve (Sig. = 0.031) and in the right TA (Sig. = 0.015) (Table 1). Levene’s test showed that all baseline variables in the two groups were homogeneous (*p* > 0.05), except for the algometry of both TA (right (Sig. = 0.042) and left (Sig. = 0.040).

### 3.2. Main Results after the Interventions

Statistically significant results were obtained in both groups for the main variables measuring pain and disability (Table 2). Pain in the NM group decreased more (−2.12 ± 1.14 points) than in the NSE group (−1.56 ± 1.53 points), obtaining highly significant results in both groups (Sig. < 0.001). In terms of disability, statistically significant results were also obtained in both groups. The NM group decreased in the NDI by 4.15 ± 2.86 points (Sig. < 0.001) and in the NSE group by 2.89 ± 3.92 points (Sig. = 0.05). The differences in both groups, both NPRS and NDI, can be seen in the box plots in Figure 2. Despite these differences, significant differences were not obtained between groups when comparing the effect of both groups (pNPRS = 0.069; pNDI = 0.094). However, a small to medium effect size of the NM group relative to the NSE group has been demonstrated for both variables, with pain being close to the medium effect size (dNPRS = 0.414; dNDI = 0.367).

Improvements were also obtained in the rest of the variables in both groups. Significant results were obtained in all the variables of ROM, strength, pain threshold to pressure, kinesophobia and catastrophization (Sig. < 0.05).

With regard to ROM, significant differences were not obtained between the two groups (Sig. < 0.05). The greater effect of the NM group stands out, with a small to medium effect size on mobility in extension and right inclination (dExtR = 0.436; dIncR = 0.457). In both variables, statistically significant results were almost achieved (pExtR = 0.059; pInclR = 0.051).

With regard to strength, significant differences were not found in any strength variable between the groups (Sig. > 0.05). However, a small to medium effect size was found in the NM group in right wrist extension (dWristExtR = 0.417). In addition, a small effect size was obtained in the NM group for right grip strength (dHGR = 0.223) and right elbow extension (dElbowExrR = 0.247).

The greatest differences between the two groups were found in the pain threshold to pressure. Statistically significant results were obtained in the pressure pain threshold in the C6 and C7 spinous processes (pPPTC6 = 0.012; pPPTC7 = 0.029) and in both TA (pPPTTAR = 0.007; pPPTTAL = 0.039). Furthermore, a mean effect size was obtained for these variables (dPPTC6 = 0.638; dPPTC7 = 0.535; dPPTTAR = 0.693; dPPTTAL = 0.501). In addition, a low to medium effect size was obtained in the NM group with respect to the NSE group on the PPT of C5 and the left median nerve (dPPTC5 = 0.406; dPPTMNL = 0.439).

Regarding catastrophization and kinesophobia, significant differences were not found between the groups (Sig. > 0.05). However, a small to medium effect size has been demonstrated in the NM group compared to the other group (dPCS = 0.364; dTAMPA = 0.276).

## 4. Discussion

### 4.1. Discussion of Results

The results have shown that both the NM group and the NSE group improve significantly in practically all variables. Significant differences were obtained in the NM group with respect to the NSE group, especially in the pressure pain thresholds (C6, C7, right and left TA), also showing a medium effect size. Significant differences were not obtained between the groups in the main variables. In both groups, improvements in pain were greater than the minimum demonstrable change in this population group. Regarding neck disability (NDI), in the NM group a mean improvement of 17.86% was obtained, very close to the MCID (19%), and in the NSE group the effect was lower (13.22%) [33]. Despite this, a low to medium effect size was demonstrated for both variables in favour of the neurodynamics group, along with neck extension and tilt mobility, right wrist extension strength, C5 pressure threshold, left median nerve and catastrophizing. A low effect size was also obtained in favour of the NM group in the variables of TAMPA, right elbow extension strength and right manual grip. In the rest of the variables, we did not obtain a sufficient effect size to make an assessment, and the NSE group did not demonstrate a sufficient effect in any of the variables to be considered clinically relevant. These results suggest that both interventions improve but that the neurodynamic intervention may have more clinically relevant results than non-specific exercise.

It may be that the effect on one variable may generate benefits on the other. If you look at our results, you can see that the ROM in extension and right tilt have been improved. These movements are the ones that most compress the nerve output [34]. The greatest improvements in strength were obtained on the right side for hand grip and wrist and elbow extension. Increased right extension and tilt movement may have indirectly improved the motor function of the nerves exiting the neck [35,36]. The improvement of mobility and the restoration of neural slippage may have had an effect on increasing the strength and pressure threshold of the affected levels.

The most encouraging results of the study are the improvement in subjective (NPRS) and objective (PPT) pain, as pain sensation and pressure pain thresholds in the neck and TA have been improved. Dry needling, massage, passive mobilisations or manipulations and even NM in patients with neuropathic pain have been shown to have positive short-term effects [37,38,39]. On the one hand, this shows a clear local analgesic effect. These results may indicate that NM can be a tool to treat pain (NPRS and PPT) due to the local analgesic effect it has shown. It also highlights the significant differences with non-specific exercise at a systemic level. This demonstrates that NM not only has local effects but can also lower pressure pain thresholds in remote body regions.

This could be one of the reasons why NM has an effect not only on neuropathic pain but also on the experience of pain in general. In fact, it has already been shown that NM treatment had a beneficial effect on pain intensity and disability in people with musculoskeletal disorders and on mechanosensitivity in asymptomatic people [32]. To our knowledge, to date there are no clinical trials that have tested NM in neck pain without neuropathic pain. In contrast, in the lumbar region NM has already been shown to show moderate effects on flexibility in healthy participants and large effects on pain and disability in people with low back pain [40]. It has also been tested in people with fibromyalgia and multiple sclerosis showing positive and potential benefits [41,42]. It is therefore possible that NM could be used as a technique in the treatment of pain in general and not only in neuropathic pain, the effect of which has already been demonstrated [18,19].

The other group also had statistically significant results in the data collected (*p* < 0.05), indicating that non-specific exercise also improves functionality and neck pain. It has been shown that mobilisation of symptomatic cervical levels is not necessary to reduce pain. Staying active and participating in exercise groups are both interesting strategies [32]. A recent meta-analysis comparing different types of exercise concluded that different types of exercise can have analgesic effects at local and remote locations, as shown by our results on the pressure pain threshold at the TA [17,43]. The results of the study agree with the current scientific literature, which shows that exercise, both specific and non-specific, can reduce neck pain in both the short and long term. In the long term, the trend seems to be that specific exercises and their performance may have a superior effect on pain reduction, although the best format is still unclear. However, the best exercise modality to reduce neck pain is still unknown [44,45]. Therefore, the nature of the exercise might be of minor importance compared to just exercising. This lack of emphasis on a specific type of exercise could be related to the numerous and widespread mechanisms through which exercise acts to alleviate pain [46].

### 4.2. Limitations

One of the aspects for which quality evidence has not yet been obtained is the long-term effect of NM; for example, osteoarthritis patients did not maintain pain reduction after three months [47], so follow-up is needed to assess the possible long-term effects of NM. Further studies should include a prospective design with follow-up. It should be noted as a limitation that the study has been monocentric and only performed in women, so that in the future it could also be extended to men and a multi-centre clinical trial could be performed.

One of the limitations of the study may have been the way the NM exercises were performed. Exercises have been governed by the sensation of tension and not by purely perfect biomechanical parameters, as this has been shown to give better results with respect to pain [48]. In addition, active NM was proposed as it did not show a significant change compared to the passive way of benefiting from the analgesic power of exercise [17,49].

### 4.3. Future Perspectives

Further research working with a larger number of patients, with different population groups, in different pathologies and assessing long-term response is essential. In relation to neck pain, it would be interesting to obtain samples of participants from other categories of neck pain, as stated in the 2017 APTA clinical practice guideline [4].

Furthermore, it is important to note that psychometric variables such as kinesiophobia and catastrophizing are implicated in chronic pain processes. These variables can be improved with well-planned exercise, but it would be interesting to include pain neuroscience education (PNE) sessions in the future [50,51]. The combination of PNE and exercise in the treatment of chronic musculoskeletal pain produces greater short-term improvements in pain, disability, kinesiophobia and catastrophic pain compared to exercise alone [52]. In addition, this joint treatment has already been shown to have very positive effects in chronic neck pain [53,54].

It has been shown that NM can achieve more clinically relevant results than non-specific exercise on pain, disability, strength, mobility, pain threshold to pressure, catastrophization and kinesiophobia in older women with chronic neck pain. NM may be one of the treatment techniques with the greatest potential for future research in this area.

## 5. Conclusions

The study shows that specific neurodynamic exercises can improve pain and disability in older women with chronic mechanical neck pain. Improvements were seen in cervical ROM, upper limb strength, pressure pain thresholds, kinesiophobia and catastrophisation. The study did not find that the effect of a specific neurodynamic exercise programme was greater than that of non-specific exercises. However, the effect size was larger in the neurodynamic group, making it more clinically relevant for treating neck pain in older women. Further research is needed to understand the long-term effects and the different population groups that have not been analysed in the study.

## Figures and Tables

**Figure 1 healthcare-12-00020-f001:**
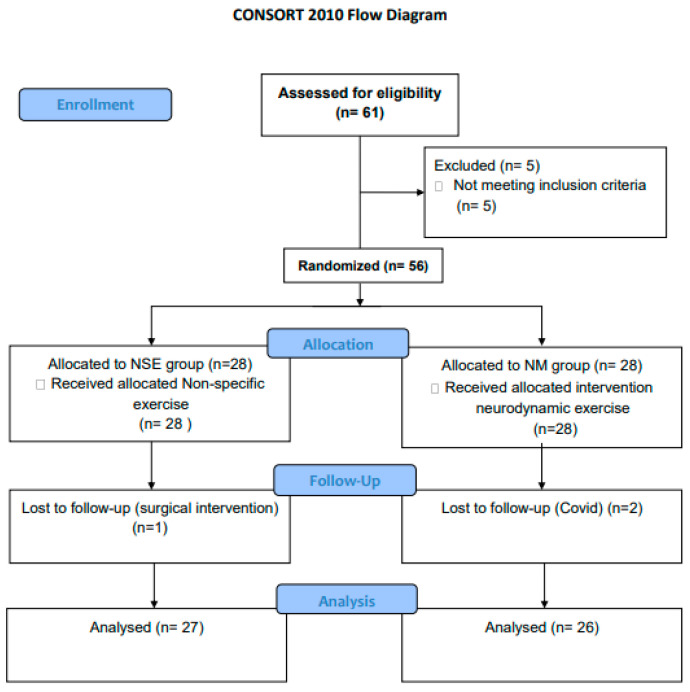
Flow diagram. Description of the course of the study from the moment of selection of participants until the analysis.

**Figure 2 healthcare-12-00020-f002:**
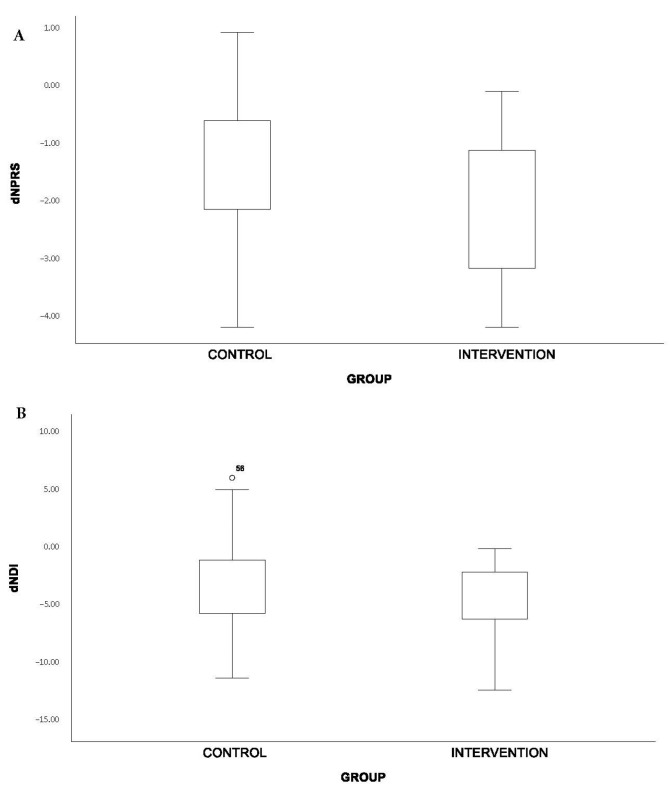
Box plots of the main variables of the study. (**A**) Box plot of differences in neck pain (NPRS) after interventions. (**B**) Box plot of differences in neck disability (NDI) after interventions.

**Table 1 healthcare-12-00020-t001:** Means and standard deviations of baseline characteristics of each group and tests for normality and homogeneity.

	NM Group (n = 28)	NSE Group (n = 28)	Shapiro–Wilk’s Test	Levene’s Test
	$\bar{x}\pm s$	$\bar{x}\pm s$	Sig.	Sig.
NPRS	5.54 ± 1.47	5.19 ± 1.5	0.212	0.703
NDI	23.23 ± 5.23	21.85 ± 4.23	0.182	0.531
ROM flex	48.73 ± 16.26	41.59 ± 11.98	0.666	0.156
ROM ext	55.42 ± 15.71	51.41 ± 14.59	0.304	0.580
ROM rot R	54.04 ± 17.60	51.44 ± 17.63	0.772	0.493
ROM rot L	56.46 ± 17.04	54.19 ± 18.74	0.550	0.310
ROM incl R	29.88 ± 9.76	31.96 ± 8.12	0.058	0.184
ROM incl L	34.23 ± 10.58	32.00 ± 9.42	0.097	0.625
HG R	18.74 ± 5.04	18.34 ± 5.04	0.909	0.754
HG L	20.13 ± 3.77	18.94 ± 3.77	0.749	0.096
D. elbow flex R	74.98 ± 22.42	65.98 ± 22.42	0.762	0.740
D. elbow flex L	73.93 ± 22.37	67.69 ± 22.37	0.107	0.652
D. wrist ext R	56.90 ± 17.08	52.20 ± 12.51	0.300	0.071
D. wrist ext L	51.273 ± 16.67	47.87 ± 14.14	0.101	0.190
D. elbow ext R	67.84 ± 22.94	55.75 ± 22.94	0.383	0.050
D. elbow ext L	64.98 ± 23.29	52.03 ± 23.29	0.084	0.378
PPT C5	2.24 ± 0.99	2.20 ± 0.74	0.370	0.338
PPT C6	2.30 ± 1.06	2.27 ± 0.73	0.723	0.138
PPT C7	2.35 ± 1.02	2.27 ± 0.86	0.064	0.586
PPT med. R	2.49 ± 0.70	2.55 ± 0.80	0.031	0.699
PPT med. L	2.60 ± 0.95	2.59 ± 0.93	0.333	0.780
PPT TA R	2.82 ± 1.01	2.76 ± 0.66	0.015	0.042
PPT TA L	2.69 ± 1.24	2.51 ± 0.75	0.360	0.040
PCS	16.54 ± 11	12.37 ± 10.20	0.010	0.539
TAMPA	27.62 ± 6.65	25.33 ± 7.39	0.660	0.664

$\bar{x}$: mean; s: standard deviation; Sig.: signification; C5: fifth cervical vertebral level; C6: sixth cervical vertebral level; C7: seventh cervical vertebral level; D: Dynamometry; ext: extension; flex: flexion; HG: hand grip; incl: inclination; L: left; med.: median nerve; NDI: neck disability index; NPRS: numeric pain rating scale; PCS: pain catastrophization scale; PPT: pain pressure threshold; R: right; ROM: range of motion; rot: rotation; TAMPA: kinesophobia’s test.

**Table 2 healthcare-12-00020-t002:** Contrast tests for comparison between pre- and post-intervention measures for each experimental group.

	NM Group (n = 26)		NSE Group (n = 27)		Student’s *t*-Test for Equality of Means	d Cohen
Variables	$\mathrm{Dif}.\;\bar{x}$	Sig.	s	Sig.	Sig.	*d*
NPRS	−2.12 ± 1.14	<0.001	−1.56 ± 1.53	0.005	0.069	0.414
NDI	−4.15 ± 2.86	<0.001	−2.89 ± 3.92	<0.001	0.094	0.367
ROM flex	−2.65 ± 14.74	<0.001	−2.89 ± 14.54	<0.001	0.477	−0.016
ROM ext	10.96 ± 9.04	<0.001	6.41 ± 11.65	<0.001	0.059	−0.436
ROM rot R	8.80 ± 6.47	<0.001	9.44 ± 14.67	0.001	0.420	0.056
ROM rot L	4.88 ± 9.80	<0.001	6.92 ± 14.21	<0.001	0.273	0.167
ROM incl R	5.61 ± 5.51	<0.001	2.37 ± 8.34	0.012	0.051	−0.457
ROM incl L	4.5 ± 7.31	<0.001	3.7 ± 8.36	0.002	0.357	−0.101
HG R	2.42 ± 2.5	<0.001	1.8 ± 2.98	<0.001	0.210	−0.223
HG L	0.76 ± 2.16	<0.001	0.42 ± 3.08	<0.001	0.319	−0.130
D. elbow flex R	9.77 ± 13.34	<0.001	9.32 ± 14.3	<0.001	0.453	−0.032
D. elbow flex L	7.42 ± 20.24	0.003	7.85 ± 13.88	<0.001	0.465	0.025
D. wrist ext R	11.83 ± 11.76	<0.001	7.45 ± 9.07	<0.001	0.068	−0.417
D. wrist ext L	7.42 ± 20.24	<0.001	7.85 ± 13.88	<0.001	0.465	00.25
D. elbow ext R	12.01 ± 14.51	<0.001	8.62 ± 12.89	<0.001	0.186	−0.247
D. elbow ext L	11.37 ± 11.33	<0.001	8.97 ± 14.55	<0.001	0.255	−0.184
PPT C5	0.75 ± 0.55	<0.001	0.53 ± 0.57	<0.001	0.073	−0.406
PPT C6	0.77 ± 0.53	<0.001	0.39 ± 0.66	<0.001	0.012	−0.638
PPT C7	0.79 ± 0.44	<0.001	0.46 ± 0.75	<0.001	0.029	−0.535
PPT med. R *	0.89 ± 0.71	<0.001	0.98 ± 4.59	0.283	0.461	0.027
PPT med. L	0.68 ± 0.79	<0.001	0.35 ± 0.72	<0.001	00.60	−0.439
PPT TA R *	0.89 ± 0.65	<0.001	0.40 ± 0.74	0.012	0.007	−0.693
PPT TA L *	0.98 ± 0.68	<0.001	0.63 ± 0.73	<0.001	0.039	−0.501
PCS *	4.08 ± 6.18	0.002	2 ± 5.2	0.032	0.096	0.364
TAMPA	−4.15 ± 5.66	<0.001	−2.48 ± 4.57	<0.001	0.121	0.276

*: non-parametric analysis; Dif. $\bar{x}$: mean difference; s: standard deviation; Sig.: signification; C5: fifth cervical vertebral level; C6: sixth cervical vertebral level; C7: seventh cervical vertebral level; *d*: effect size; D: Dynamometry; ext: extension; flex: flexion; HG: hand grip; incl: inclination; L: left; med.: median nerve; NDI: neck disability index; NPRS: numeric pain rating scale; PCS: pain catastrophization scale;; PPT: pain pressure threshold; R: right; ROM: range of motion; rot: rotation; TAMPA: kinesophobia’s test.

## Data Availability

The data presented in this study are available upon request from the corresponding author. The data are not publicly available due to compliance with data protection regulations.

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
