# Peer review of "Specific Neurodynamic Exercises on Pain and Disability in Old Women with Chronic Mechanical Neck Pain: A Randomized Controlled Trial"

_healthcare, 2023, doi:10.3390/healthcare12010020_

Round 1

Reviewer 1 Report

Comments and Suggestions for Authors

First of all, congratulations on your work and then I would like to make a number of suggestions to improve your manuscript.

The introduction is well referenced and up to date. I would perhaps develop the section on risk factors further.

In material and methods, you use VAS, but you do not specify whether this value is collected only once, whether it is the average or maximum intensity reported over a period of time.....

In the results section, have you checked if there are differences in the intention-to-treat or per-protocol analyses you have included? If not, include that analysis and assess whether there are differences between the two. On the other hand, it would be more visual for the reader to include box plots to facilitate the analysis of the results.

In the discussion section, I recommend commenting on the results with the minimal clinically important differences and the minimal detectable differences for all variables. On the other hand, I recommend further discussing the importance of education plus exercise in the treatment of neck pain and comparing it with other interventions that combine both tools. There are previous meta-analyses that discuss this construct and other methods that combine these tools such as the famous back schools that are commonly used in this field and that would be interesting to discuss.

Congratulations again on your results and work and I hope that these comments will help to improve your manuscript.

Reviewer 2 Report

Comments and Suggestions for Authors

Dear Authors

Thanks a lot for the opportunity you have offered me to revise the fascinating manuscript “Specific neurodynamic exercises on pain and disability in old women with chronic mechanical neck pain”.

As a significant strength, this manuscript evaluates the effect of neurodynamic exercises on disability and neck pain in elderly women during four weeks. This proposal is a novelty in the field and adds information to the existing evidence in the literature produced in the field.

As a major weakness, the manuscript sometimes needs few details and clarity concerning methodological steps that would help improve the understanding of the manuscript.

Overall, the paper is well-structured, developed and written. Thus, my peer review is a minor revision. After integrating the improvements, I will be happy to accept it.

§MINOR ISSUES

#ABSTRACT:

*”was conducted”: prospectively? Please be more specific.

#INTRODUCTION

*background: the introduction is well-written and structured. Congratulations. One dimension that could be further improved and integrated concerns the subjective experience that neck pain patients have of their disability. In this regard, I suggest the authors read and integrate this qualitative systematic review (doi: 10.1093/ptj/pzac080 PMID: 35708498) to improve the quality of their manuscript.

#RESULTS:

*formatting: I suggest authors organise their results into chapters following CONSORT.

*abbreviations: please check that all abbreviations are in full in the manuscript.

#DISCUSSION:

*limitations: I suggest reporting as limitations that the study is monocentric, has no long-term follow-up, and that it only includes women.

§MAJOR ISSUES

#METHODS:

*CONSORT: I suggest the authors organise the methods into chapters following CONSORT.

*CONSORT: I suggest the authors include the CONSORT in the appendix, compile it and add the reference in the manuscript.

*TIDIER: I suggest the authors report the description of the interventions and the checklist using the TIDIER checklist. This will improve the quality of the manuscript (https://www.equator-network.org/reporting-guidelines/tidier/)

*Outcomes: for each outcome, report a bibliographic reference. Pain and range of motion are missing. So I suggest you include these references for neck pain (doi: 10.2519/jospt.2009.2930 PMID: 19521015) and for neck ROM (doi: 10.1016/j.jmpt.2017.07.002 PMID: 29187311).

*Interventions and control: are they evidence-based? Please add references for both.

*statistical analysis: was it per protocol or intention-to-treat?

*Ethics: is the Clinical trial registration prospective or retrospective? Please be precise.

Comments on the Quality of English Language

The english is good. Only minor revisions are needed.
